

# The Arctic overturning circulation: transformations, pathways and timescales

Jakob Dörr[1*], Carlo Mans[1*], Marius Årthun[1], Kristofer Döös[2], Dafydd Gwyn Evans[3], and Yanchun He[4]

[1]Geophysical Institute, University of Bergen and Bjerknes Centre for Climate Research, Bergen, Norway
[2]Department of Meteorology, Stockholm University, Stockholm, Sweden
[3]National Oceanography Centre, Southampton, UK
[4]Nansen Environmental and Remote Sensing Center and Bjerknes Centre for Climate Research, Bergen, Norway
[*]These authors contributed equally to the paper

**Correspondence:** Jakob Dörr (jakob.dorr@uib.no)

**Abstract.**

   The Arctic is the northernmost terminus of the Atlantic Meridional Overturning Circulation and is an important source of the densest waters feeding its lower limb. However, relatively little is known about the structure and timescales of the Arctic overturning circulation, and which pathways contribute most to the transformation of Atlantic Waters into dense waters and Polar

Waters. In this work, we combine a Eulerian water mass transformation framework and Lagrangian tracking to decompose the time-mean Arctic overturning circulation in an eddy-rich (1/12˚) global ocean hindcast (1979-2015). We show that the Atlantic Water branch through the Barents Sea dominates dense Arctic overturning, and that a large portion of these transformed waters takes many decades to exit Fram Strait. Furthermore, we show that surface forcing in the Barents Sea and north of Svalbard dominates dense overturning, but local subsurface mixing with shelf waters and between the two Atlantic Water branches plays

an important role for the Fram Strait branch. Our work identifies the dominant processes of the Arctic overturning circulation, and contributes to understanding its future changes and their impact on the stability of northern overturning.

## 1   Introduction

A key component of the Atlantic Meridional Overturning Circulation (AMOC) is the transformation of Atlantic Waters into dense waters at high latitudes (Buckley et al., 2023). Considerable attention has been directed toward understanding the over-

turning processes in the subpolar North Atlantic and Nordic Seas, where the production of dense waters supply the lower limb of the AMOC (Dickson and Brown, 1994; Chafik and Rossby, 2019; Petit et al., 2020; Yeager et al., 2021; Årthun, 2023). Recently, it has been recognized that also the Arctic Ocean is a major component in the production of the densest waters sustaining the AMOC (Zhang and Thomas, 2021). However, the understanding of the overturning circulation in the Arctic Ocean is much less developed and a detailed description of overturning processes in the Arctic is lacking.

Warm and salty Atlantic Water enters the Nordic Seas across the Greenland-Scotland Ridge and flows northward along its eastern boundary (Orvik and Niiler, 2002). It then enters the Arctic Ocean as two distinct branches, via the Fram Strait and the Barents Sea opening (Figure 1; Beszczynska-Möller et al., 2011). Within the Arctic Ocean, the Fram Strait and Barents



Sea branches of Atlantic Water merge at St. Anna trough, and continue flowing around the deep Arctic basins as a cyclonic boundary current (Coachman and Barnes, 1963; Timmermans and Marshall, 2020). As the two branches meet, they interact

and exchange properties (Schauer et al., 2002). As Atlantic Water circumnavigates the Arctic Ocean, it is both cooled and freshened through heat loss to the atmosphere as well as interaction with sea ice and shelf waters (Rudels et al., 2015; Ivanov et al., 2024). A large part of the surface heat loss occurs in the Barents Sea (Skagseth et al., 2020; Smedsrud et al., 2013). As a result, the Barents Sea branch is colder and denser than the Fram Strait branch, and occupies a deeper part of the water column than the Fram Strait branch (Aksenov et al., 2011; Moat et al., 2014). On the Pacific side of the Arctic Ocean, inflowing Pacific

Waters undergo comparable modification processes, and serve as an additional source of heat and freshwater (Tsubouchi et al., 2024).

The transformation of these inflowing water masses gives rise to the production of two key outflow products: Dense Waters and Polar Waters (Haine, 2021; Tsubouchi et al., 2024). These outflow types are frequently characterized as resulting from two distinct circulations in a so-called double estuarine representation of the Arctic Ocean circulation (Carmack and Wassmann,

2006; Eldevik and Nilsen, 2013; Haine, 2021; Brown, 2019). The thermal overturning branch represents the conversion of Atlantic Water into cooler and slightly fresher waters, resulting in the formation of Dense Water. These dense waters recirculate back to the Nordic Seas through the deep sections of the Fram Strait. In contrast, the estuarine branch represents the gradual freshening and cooling of Atlantic Water through mixing with fresher waters, formed largely through ice melt and river runoff, to produce buoyant Polar Waters, which are exported through outflows in the upper layers of the Davis and Fram Straits (e.g.,

Rudels et al. (2004)).

In a warming climate, the overturning circulation in the North Atlantic (i.e., the AMOC) is projected to decline as the necessary dense water formation is inhibited by warmer and fresher surface waters (Asbjørnsen and Årthun, 2023; Weijer et al., 2020). In contrast, in the Arctic Ocean, sea-ice loss might lead to a stronger surface exposure of Atlantic Water and hence increased dense water formation, potentially stabilizing the northern overturning circulation (Lique and Thomas, 2018;

Bretones et al., 2022; Årthun et al., 2025). In order to better project the response of Arctic overturning to future warming, it is important to understand how Atlantic Water entering the Arctic Ocean is transformed before being exported as dense waters to join the lower limb of the the AMOC. Although the general transformation processes required to produce the outflow products are known, the relative contributions of surface forcing and interior mixing are not. It is also not established nor quantified where along the Atlantic Water pathways the transformation is most pronounced, or which processes dominate. While it is

known that the Barents Sea branch undergoes the strongest surface heat loss (Skagseth et al., 2020), the relative importance of the two Atlantic Water branches for the thermal branch of Arctic overturning has not been quantified.

In this work, we analyze the mean structure of the Arctic overturning circulation in a global eddy-rich ocean hindcast, using both an Eulerian water mass transformation framework (Walin, 1982; Evans et al., 2023) and offline Lagrangian trajectories. We quantify the transformation of Atlantic Water into Polar Water and dense waters, and estimate the relative contribution of

surface processes and internal mixing, and their seasonal variability. Furthermore, we quantify the relative importance of the two Atlantic Water branches, and the geographic locations of water mass transformation.





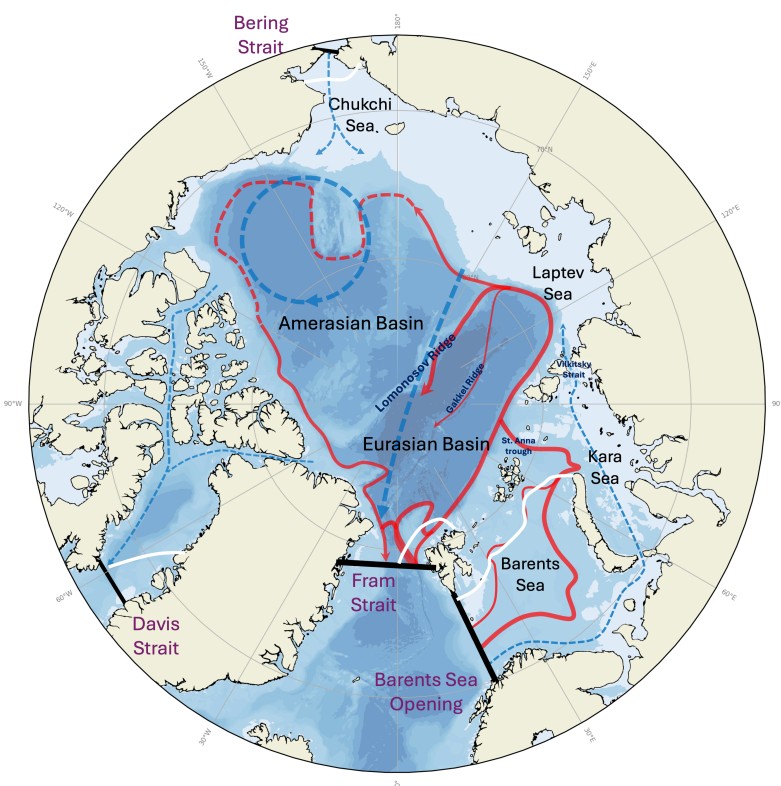

**Figure 1.** Overview of the Arctic Ocean, including its four oceanic gateways, relevant regional seas, ocean basins and ridges, and a schematic of the major Atlantic Water (red) and surface layer (blue) pathways. White lines denote approximate locations of the annual mean sea-ice edge.

## 2 Materials and Methods

### 2.1 NEMO ocean hindcast

For our analysis of the Arctic overturning circulation, we use output from a global NEMO ocean hindcast simulation (ORCA0083-N06; Moat et al., 2016). The simulation is based on NEMO-LIM2 with a horizontal resolution of 1/12°, which is approximately 3 - 5 km in the Arctic, and 75 unevenly spaced depth layers. The model is forced with atmospheric and river runoff data from the DRAKKAR forcing set 5.2 (Dussin et al., 2016) and was run from 1958 to 2015. To prevent excessive drifts in salinity, sea-surface salinity is relaxed towards climatology (Moat et al., 2016). We consider output from 1979 to 2015 in this study. We make use of the monthly mean velocity and tracer output.

ORCA0083-N06 has previously been used to study the Arctic Ocean, and the upper ocean circulation, sea surface height and mixed-layer depth has been found to match well with observations (Kelly et al., 2018, 2019; Wilson et al., 2021). Furthermore, NEMO has been extensively validated in the Arctic Ocean, and previous and similar versions have found to generally perform



**Table 1.** Comparison of ocean volume transports from ORCA0083-N06 with observational estimates. Uncertainty values denote the interannual standard deviation over 1979-2015 for the model.

| Gateway | ORCA0083-N06 (1979–2015) | Observation-based estimate |
|---|---|---|
| Bering Strait | $1.3 \pm 0.5$ Sv | $1.0 \pm 0.5$ Sv (Woodgate, 2018) |
| Barents Sea Opening | $3.5 \pm 1.0$ Sv | 3.3 Sv (Rudels et al., 2015) |
| Fram Strait | $-3.0 \pm 0.9$ Sv | $-2.0 \pm 2.7$ Sv (Beszczynska-Möller et al., 2011) |
| Davis Strait | $-1.8 \pm 0.7$ Sv | $-2.6 \pm 1.0$ Sv (Curry et al., 2011) |

well (Lique et al., 2010). Nevertheless, we evaluate the model against observational estimates of net volume transports at the gateways in Table 1. Acknowledging that the observations cover different periods than the model, the net volume transports
at the Arctic gateways are generally consistent with observational estimates, although ORCA0083-N06 overestimates the net outflow through Fram Strait and underestimates the outflow through Davis Strait compared to the most recent estimates. However, the inflow of Atlantic water through Fram Strait (T > 2°C, S > 34.7) for the period 1997 - 2015 is 3.0 Sv, close to recent estimates of $3.0 \pm 0.2$ Sv (Beszczynska-Möller et al., 2012). The inflow of Atlantic Water (T > 3°C) through the Barents Sea Opening between 75°N and 73.5°N is 2.4 Sv, consistent with estimates from mooring data of $2.1 \pm 1.0$ Sv (Smedsrud
et al., 2022). The temperature and depth of the Atlantic Water core in the Arctic Ocean is similar to estimates from the PHC3.0 climatology (Steele et al., 2001), but with a cold bias in the Makarov Basin (Fig. A1). We conduct a more thorough evaluation of the Arctic overturning streamfunction and Atlantic Water pathways below, and show that the hindcast is able to realistically simulate both.

## 2.2 Water mass transformation framework

Following previous studies, we use the Walin framework (Walin, 1982) to diagnose the volume and buoyancy budget for the Arctic Ocean in density ($\sigma$) and and temperature-salinity ($T - S$) space. Here we define the Arctic Ocean domain by its four major gateways: Fram Strait, the Barents Sea Opening, Bering Strait, and Davis Strait (Figure 1). The volume tendency in a given density class can be expressed as the sum of the net advective transport across the boundaries of that class and the divergence of water mass transformation within it (Walin, 1982; Buckley et al., 2023; Evans et al., 2023; Zou et al., 2024):

$$\frac{dV_\sigma(\sigma^*, t)}{dt} = M_\sigma(\sigma^*, t) + \frac{\partial G_\sigma(\sigma^*, t)}{\partial \sigma}, \tag{1}$$

where $V_\sigma(\sigma^*, t)$ is the volume of water, $M_\sigma(\sigma^*, t)$ represents the net advective transport and $G_\sigma(\sigma^*, t)$ denotes the water mass transformation within a tracer bin centered on density $\sigma_0^*$, and bounded by $\sigma_0^* \pm \Delta\sigma_0/2$. As such, the convergence of the water mass transformation is also referred to as the water mass formation (WMF), and divergence is referred to as water mass destruction. The water mass transformation processes that affect water with density $\sigma_0^*$ can be summarized as air-sea buoyancy





fluxes where $\sigma_0^*$ outcrops, and unresolved residual interior mixing processes, such that:

$$G_\sigma(\sigma^*,t) = F_\sigma^{sfc}(\sigma^*,t) + F_\sigma^{res}(\sigma^*,t), \tag{2}$$

with $F_\sigma^{sfc}(\sigma^*,t)$ the surface forced component and $F_\sigma^{res}(\sigma^*,t)$ the residual.

Similarly, the volume tendency in a given $T - S$ class can be expressed as the sum of the net advective transport across the boundaries of that class and the divergence of water mass transformation within it:

$$\frac{dV_{S\Theta}(\Theta^*,S^*,t)}{dt} = M_{S\Theta}(\Theta^*,S^*,t) + \frac{\partial G_{S\Theta}(\Theta^*,S^*,t)}{\partial \Theta} + \frac{\partial G_{S\Theta}(\Theta^*,S^*,t)}{\partial S}, \tag{3}$$

where $V_{S\Theta}(\Theta^*,S^*,t)$ is the volume of water within a tracer bin centered on temperature $\Theta^*$ and salinity $S^*$, and bounded by $\Theta^* \pm \Delta\Theta/2$ and $S^* \pm \Delta S/2$. $M_{S\Theta}(\Theta^*,S^*,t)$ represents the net advective transport and $G_{S\Theta}(\Theta^*,S^*,t)$ denotes the water mass transformation in the domain. The net advective transport is defined as:

$$M_{S\Theta}(\Theta^*,S^*,t) = \psi_{S\Theta|\text{Fram}} + \psi_{S\Theta|\text{BSO}} + \psi_{S\Theta|\text{Bering}} + \psi_{S\Theta|\text{Davis}}, \tag{4}$$

where $\psi$ is the net volume transport within a tracer bin along a section and is calculated as

$$\psi_{S\Theta}(\Theta^*,S^*,t) = \iint \Pi[\Theta,\Theta^*]\,\Pi[S,S^*]\,\nu\,dx\,dz.$$

Here, $\Pi$ is a boxcar function equal to 1 if $\Theta \in [\Theta^* - \Delta\Theta/2, \Theta^* + \Delta\Theta/2]$ (and similarly for salinity), and 0 otherwise, while $v$ is the velocity normal to the section (positive northward). We decompose the water mass transformation terms into separate terms as:

$$G_\Theta(\Theta^*,S^*,t) = F_\Theta^{sfc}(\Theta^*,S^*,t) + F_\Theta^{res}(\Theta^*,S^*,t), \qquad G_S(\Theta^*,S^*,t) = F_S^{sfc}(\Theta^*,S^*,t) + F_S^{res}(\Theta^*,S^*,t), \tag{5}$$

where $F_\Theta^{sfc}$ and $F_S^{sfc}$ are referred to as the surface forced transformation. The surface forced transformations are given by:

$$F_\Theta^{sfc}(\Theta^*,S^*,t) = \int \frac{\Pi(\Theta,\Theta^*)}{\Delta\Theta}\Pi(S,S^*)\frac{Q_{\text{net}}}{\rho C_p}\,dA, \tag{6}$$

$$F_S^{sfc}(\Theta^*,S^*,t) = \int \Pi(\Theta,\Theta^*)\frac{\Pi(S,S^*)}{\Delta S}f_{\text{net}}\,dA, \tag{7}$$

where $Q_{\text{net}}$ is the net surface heat flux, $f_{\text{net}}$ is the net surface freshwater flux (including evaporation, precipitation, sea ice melting/freezing and river runoff), $\rho$ is seawater density at the surface, and $C_p$ is the specific heat capacity of seawater.

The volume $V_{S\Theta}$ can be calculated from the temperature and salinity fields within the Arctic Ocean domain, while the net advective transport $M_{S\Theta}$ is calculated using velocity, temperature, and salinity across the four major Arctic gateways. We use a bin spacing of $\Delta\sigma_0 = 0.025$ kg m$^{-3}$ and following Evans et al. (2014), we use a resolution in $T - S$ space such that the contribution of changes in $T$ and $S$ are approximately similar in density space: $\Delta\Theta = 0.1\,°C$ and $\Delta S = 0.01$ g kg$^{-1}$. From the volume change and boundary fluxes, the water mass transformation divergence $G_\Theta$, $G_S$ is calculated using the inverse





methods introduced in Evans et al. (2014). The surface forced transformations $F_{\Theta}^{sfc}$ and $F_{S}^{sfc}$ are then computed using net surface heat flux and freshwater flux, respectively. $F_{\Theta}^{res}$ and $F_{S}^{res}$ are computed as residuals after calculating all other terms in Equation 3. These residuals are commonly interpreted as the contributions from interior mixing along temperature and salinity

coordinates, though they may also include other unresolved processes, including sea-surface restoring (Buckley et al., 2023; Evans et al., 2023; Zou et al., 2024).

## 2.3    Lagrangian tracking

To analyze the pathways, timescales, and locations of water mass transformation of the Arctic overturning circulation in the ORCA0083-N06 hindcast, we trace waters entering the Arctic Ocean through the four gateways using TRACMASS version 7.0

(Döös et al., 2017; Aldama-Campino et al., 2020). TRACMASS solves the trajectories of parcels through each model grid cell analytically by assuming that the velocity field varies linearly in time (between output time steps) and space (between opposite grid cell walls). An important property of the algorithm is that it conserves mass, or in the case of NEMO, volume, such that each trajectory retains a constant volume transport throughout the tracking. This makes it possible to calculate Lagrangian streamfunctions, and thereby decompose the total Eulerian Arctic overturning streamfunction into contributions from different

pathways (Berglund et al., 2023; Tooth et al., 2024).

For the Lagrangian tracing experiment, we calculate the average annual cycle over 1979–2015. We start trajectories at all grid cells across the four Arctic gateways every month. To obtain a sufficient resolution, we assign each trajectory a maximum volume transport of 2500 m$^3$/s. If the volume transport in a model grid cell exceeds this value, multiple trajectories are started in that grid cell. In total, around 150.000 trajectories are started. We track the trajectories forward in time using the climato-

logical monthly velocity and tracer fields. Trajectories are terminated either if they exit the Arctic by returning to the inflow gateways, or after 500 years. We calculate Lagrangian streamfunctions following Tooth et al. (2024). To quantify water mass transformation along the pathways, we calculate the Lagrangian mass, heat and salt divergence for each grid box, following Berglund et al. (2017) and Dey et al. (2024).

## 3    Results

### 3.1    Overturning in density-space

We will first quantify the time-mean water mass transformation for the Arctic Ocean, with a particular focus on the overturning circulation. We start by considering the overturning streamfunction in density ($\sigma$) space (Figure 2a), obtained by integrating $M(\sigma^*)$ over $\sigma$. The overturning streamfunction has a local minimum and a local maximum, indicating the presence of both a negative and a positive overturning cell, which are referred to as the estuarine and thermal overturning cells, respectively

(Haine, 2021). The estuarine cell has a minimum of $-0.96$ Sv at $\sigma_0 = 27.0$ kg m$^{-3}$, while the thermal cell reaches a maximum of 3.09 Sv at $\sigma_0 = 27.95$ kg m$^{-3}$. The Arctic Ocean thus densifies inflowing waters at a rate approximately three times greater than it lightens them. Compared to the observation-based estimate of Tsubouchi et al. (2024), the thermal cell is slightly





stronger but agrees well in density structure, whereas the estuarine cell is weaker and peaks at a lower density, as well as extending through a much lower density level of approximately $\sigma_0 = 25.0$ kg m$^{-3}$. Finally, Tsubouchi et al. (2024) find a third

cell, which represents the transformation of Pacific Waters to a denser outflow. This third cell is likely obscured in the hindcast by the large amounts of export of waters between $\sigma_0 = 25.0$ kg m$^{-3}$ and $\sigma_0 = 26.5$ kg m$^{-3}$. Overall, however, there is good agreement between ORCA0083-N06 and that inferred from available observations.

The volume tendency $\frac{dV_\sigma}{dt}$ in the Arctic Ocean is generally small, except for $\sigma_0 > 27.75 - 28.0$ kg m$^{-3}$, where there is net water mass formation (Figure 2b), and $\sigma_0 > 28.0 - 28.1$ kg m$^{-3}$, where there is net water mass destruction. This hints at a

long term trend of buoyancy gain in the Arctic Ocean deep waters. The diapycnal transformation due to surface buoyancy fluxes, $F_\sigma^{sfc}$, is directed toward higher densities across nearly all density classes and reaches a maximum of -2.32 Sv at $\sigma_0 = 27.75$ kg m$^{-3}$ (Figure 2a), peaking at a lower density than the overturning. Surface forcing contributes to the destruction of water masses lighter than $\sigma_0 < 27.75$ kg m$^{-3}$, and to the production of water masses in the range $\sigma_0 = 27.75$–28.1 kg m$^{-3}$. Notably, it contributes substantially to the formation of dense waters around $\sigma_0 = 28.0$ kg m$^{-3}$, which are exported from the

Arctic as part of the dense outflow, suggesting that surface buoyancy forcing is most important for the thermal cell.

The residual transformation $F_\sigma^{res}$ (interpreted as mixing) shows a more complex structure (Figure 2b). At lower densities, specifically below $\sigma_0 = 27.7$ kg m$^{-3}$, mixing drives water mass formation with the characteristics of lighter surface polar waters, supplying the majority of the waters exported as part of the estuarine circulation cell. Mixing then acts destructively over a broad intermediate density range, with pronounced water mass destruction between $\sigma_0 = 27.7$ and 27.95 kg m$^{-3}$ (Figure 2b),

corresponding to a part of the Atlantic Water inflow. The lighter fraction of this destruction contributes to the estuarine cell by mixing Atlantic Water upward into fresher surface layers, while the denser fraction, along with the destruction observed at the upper end of the density range for $\sigma_0 > 28.05$ kg m$^{-3}$, leads to a net convergence of transformation around $\sigma_0 = 28.0$– 28.05 kg m$^{-3}$. This convergence corresponds to the majority of dense water mass exports, suggesting that while surface fluxes can provide much of the required densification to sustain the Arctic overturning circulation, preconditioning the inflowing

waters, mixing sets the final properties of the dense waters that are exported from the Arctic Ocean through its deeper branches, analogous to the findings of Evans et al. (2023) for the formation of North Atlantic Deep Water.

## 3.2 Overturning in temperature–salinity space

While the water mass transformation in density space offers general insights into the transformations required to sustain the Arctic overturning circulation, a diagnosis of the water mass transformation in temperature-salinity $(T - S)$ space permits a

more detailed, process based analysis. The time-mean volume distribution of the Arctic Ocean $T - S$ as well as some of the key water masses are visualized in Figure 3. In this section, as before, we place a special focus on the thermal branch of the Arctic overturning circulation and the water mass transformation processes that support it. We will start by considering the time-mean overturning divergence $M_{S\Theta}$ (Figure 4, colors) and the water mass transformation $G_{S\Theta}$ (Figure 4, vectors) required to support it. The use of thermohaline coordinates reveals the presence of the Pacific cell (Tsubouchi et al., 2024), where relatively warm

and fresh Bering Strait inflow ($\Theta = -2 - 7\ ^\circ$C and $S = 32 - 33$ g kg$^{-1}$), undergoes a predominantly cooling transformation shown by the vectors in Figure 4. This cell's presence is obscured in $\sigma$-space (Figure 2).





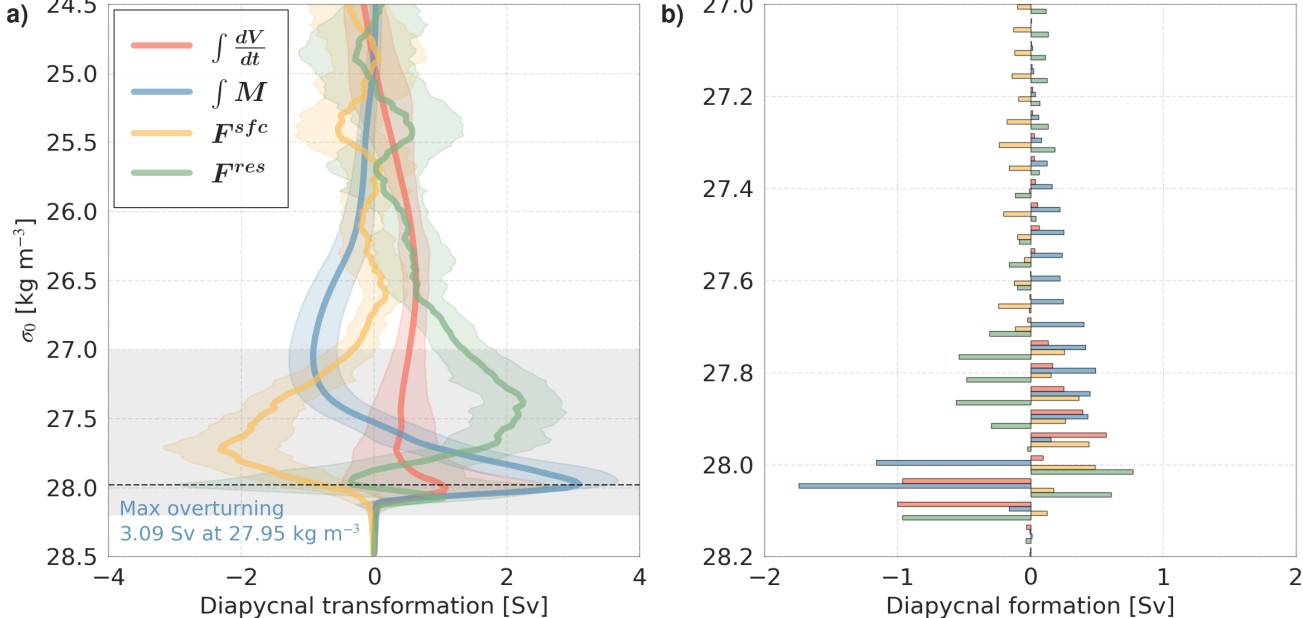

**Figure 2.** Time-mean diapycnal water mass transformation (a) and formation (b) for the Arctic Ocean from 1979-2015. The respective colors represent the integrated volume tendency (red), the overturning streamfunction (blue), the surface forced transformation (yellow) and the residual transformation (green), as well as their yearly standard deviation in shading. The quantities in (b) are calculated as the divergence of those in (a). The gray shading in (a) represents the extent of the y-axis in (b).

We first turn our focus to the estuarine circulation cell, which governs the net freshwater export from the Arctic Ocean (Rudels and Carmack, 2022). In contrast to the comparatively small Pacific cell, the estuarine cell exhibits a more persistent structure in both $\sigma$ space and $T$–$S$ space. In $T$–$S$ space, however, it is characterized by two distinct branches (visible as two blue curves converging on the freezing line in Figure 4), corresponding to different Arctic outflow pathways. Both branches export Polar Waters, defined here by near-freezing temperatures and relatively low salinities ($S = 31$–$34$ g kg$^{-1}$). While their exported properties are broadly similar, the branches differ in their source waters and transformation pathways.

The warmer branch originates primarily from Atlantic Water inflows ($S > 34.5$ g kg$^{-1}$, $\Theta > 5$ °C) entering through the Barents Sea Opening and Davis Strait (not shown). In $T$–$S$ space, the pathway of this branch reflects progressive modification of warm, saline Atlantic inflows through cooling and dilution by fresher surface waters, sea-ice melt, and runoff. This results in outflowing Polar Waters of approximately ($S \approx 31$–$33$ g kg$^{-1}$). The colder branch is fed predominantly by denser and cooler Atlantic Waters ($S > 35$ g kg$^{-1}$, $\Theta > 0$°C). These waters undergo substantial modification through mixing with fresher local waters as well as freshwater inputs from ice melt, producing an outflow that likewise emerges as Polar Water but with slightly higher maximum salinities ($S \approx 34$ g kg$^{-1}$).



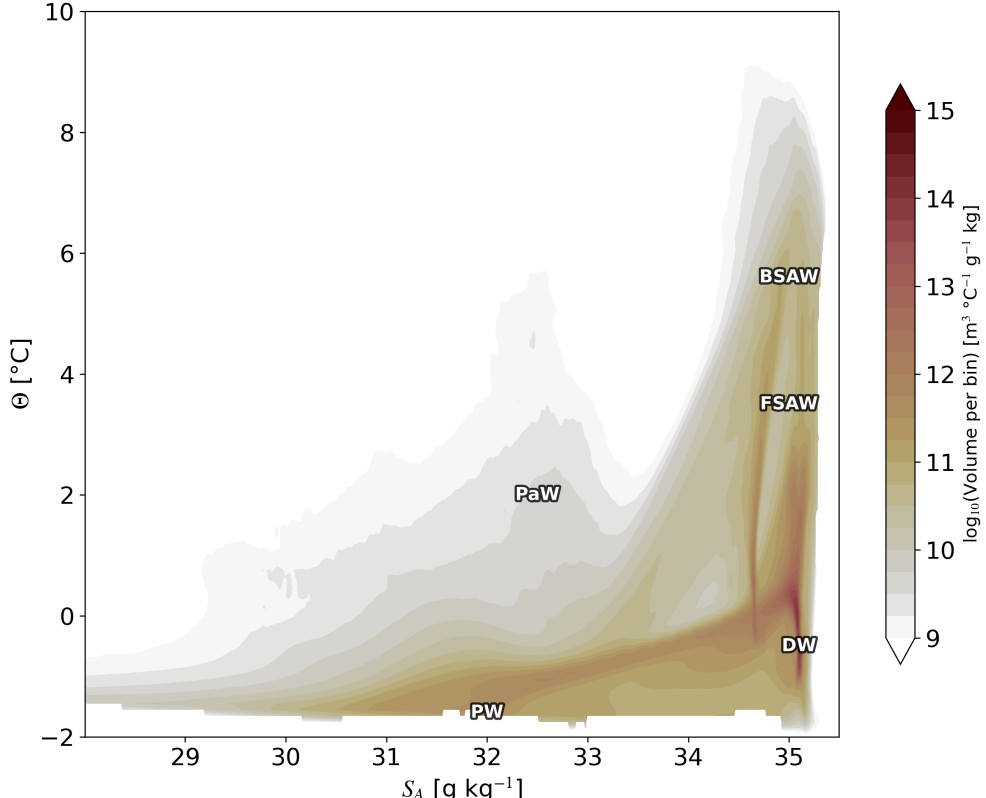

**Figure 3.** Time-mean volumetric $T - S$ distribution in the Arctic Ocean. The major water masses are marked by their abbreviations: Barents Sea Atlantic Water (BSAW), Fram Strait Atlantic Water (FSAW), Pacific Water (PaW), Dense Water (DW) and Polar Water (PW).

Next, we consider the thermal overturning cell, which occupies a relatively narrow density range ($\sigma = 27.5 - 28.2$), yet dominates the Arctic's volume transport. This cell is primarily driven by the inflow of warm, saline Atlantic waters ($\Theta > 2°$C and $S > 35$ g kg$^{-1}$), which separate into two distinct branches in T-S space: the denser, cooler, and slightly fresher Fram Strait Branch ($\Theta = 2 - 5°$C) and the lighter, warmer ($\Theta > 4$ °C), and saltier Barents Sea Branch. The water mass transformation, $G_{S\Theta}$, highlights the transformation from inflow to outflow properties (Figure 4, vectors). While both branches undergo net

cooling with slight freshening, the warmer waters entering as part of the Barents Sea Branch experience the most substantial transformation, ultimately converging toward the cold, dense Fram Strait outflow ($-1 < \Theta < 1$ °C, $35.0 < S < 35.1$ g kg$^{-1}$). This outflow forms the primary deep limb of the Arctic overturning circulation.

As the thermal cell dominates volume transports and concerns mainly inflowing Atlantic Water and densified outflows of similar salinity, it is instructive for a process based analysis to limit our focus to the specific transformation processes in

the range of T and S values corresponding to the thermal overturning cell. To that end, we consider the components of the decomposed WMT: the surface forced WMT $F_{S\Theta}^{sfc}$ and the residual WMT $F_{S\Theta}^{res}$ (Equation 5, Figure 5). The decomposition



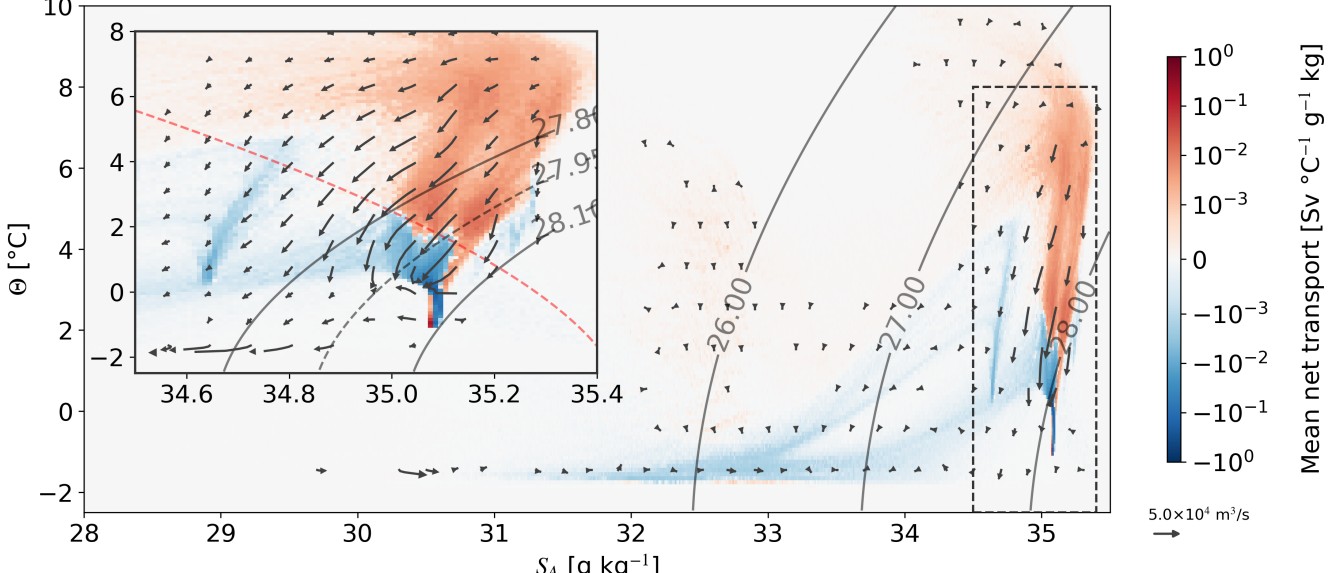

**Figure 4.** Time-mean diathermohaline net advective transport $M_{S\Theta}$ (colors) and water mass transformation $G_{S\Theta}$ (vectors) for the Arctic Ocean from 1979-2015. The colors represent net inflow (red) and net outflow (blue) per T-S bin. The arrows represent the combined transformation in both diathermal (vertical) and diahaline (horizontal) directions. The inset focuses on the thermal overturning cell's $T - S$ ranges. The dashed contours represent the density (black) and spiciness (red) of maximum overturning strength.

reveals that in the time-mean, the Atlantic Water inflowing as part of the Barents Sea Branch ($S > 35$ g kg$^{-1}$, $\Theta > 4°$C), undergo substantial cooling due to air-sea heat fluxes. Surface cooling is the dominant process until they reach approximately 2.5°C, below which ice melt becomes increasingly important, indicated by the transformation vectors bending towards lower
salinities due to the addition of freshwater (Figure 5a). This transformation results in the highly effective destruction of large volumes of warm inflow waters, consistent with a large surface forced diapycnal transformation in Figure 2. As such, the Barents Sea expectedly dominates the surface forced component. Conversely, the Fram Strait Branch is cooled to a lesser extent, though both branches ultimately contribute to the formation of substantial volumes of dense waters ($\sigma > 27.75$ kg m$^{-3}$) spanning a broad range of $T - S$ characteristics. We further see both strong formation and destruction of waters near
the freezing point, which is related to the seasonal melting and freezing cycle. Finally, surface forcing is responsible for the production of the densest waters found in the Arctic: the highly saline, near freezing point Barents Sea dense waters (Årthun et al., 2011).

     The residual component, on the other hand, is much more clearly associated with a net freshening of the inflowing salty Atlantic Water (Figure 5b). Here we interpret the residual component as the contribution from ocean mixing, but make no
explicit distinction between vertical mixing or horizontal mixing. However, the orientation of the thermohaline transformation vectors with respect to the isopycnals drawn in Figure 5b leaves clues as to the nature of the relevant mixing processes.





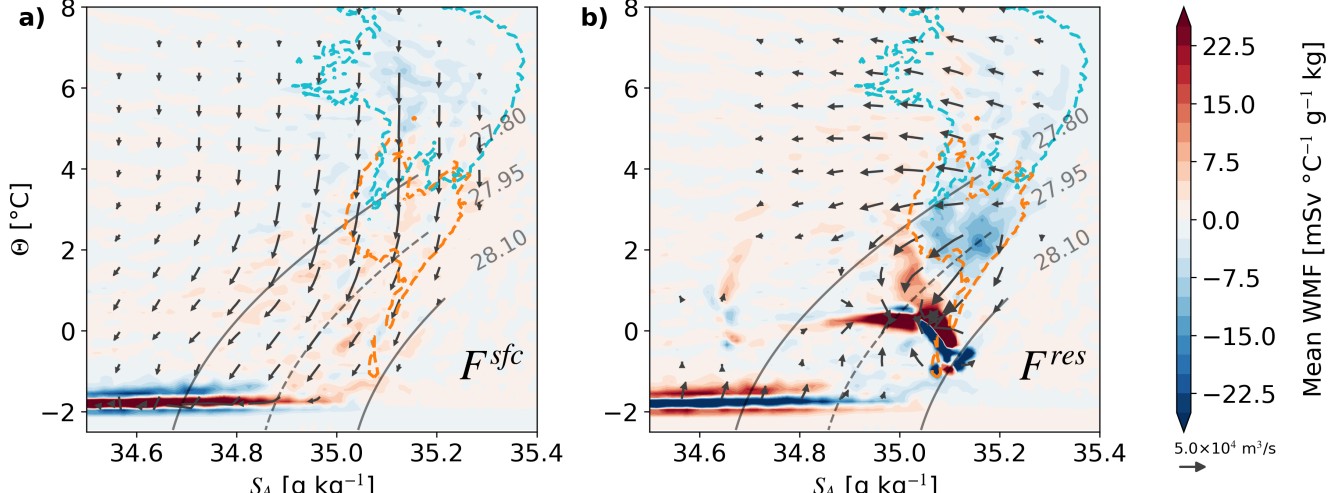

**Figure 5.** Time-mean surface forced (a) and residual components (b) of the diathermohaline water mass transformation for the Arctic Ocean from 1979-2015. The colors represent water mass formation (red) and water mass destruction (blue). The arrows represent the combined transformation in both diathermal (vertical) and diahaline (horizontal) directions. The dashed contours represent the density of maximum overturning strength (black), as well as the approximate Fram Strait (orange) and Barents Sea (cyan) inflow ($M_{S\Theta} > 2$ mSv °C$^{-1}$ g$^{-1}$ kg at the respective gateways).

Transformation vectors aligned parallel to isopycnals imply purely isopycnal mixing, while those oriented perpendicular to isopycnals are indicative of diapycnal mixing.

Firstly, mixing leads to a seasonal transformation near the freezing point that opposes the effect of surface forcing. These
opposing effects largely cancel, resulting in little net impact of much of the melting and freezing cycle on the Arctic WMT.
Meanwhile, both inflow branches experience freshening, but there is more destruction of waters that have properties associated
with the Fram Strait Branch. Notably, there is substantial net destruction of waters with temperature characteristics ($\Theta =$
$1.5-3$°C) corresponding to Fram Strait inflow waters. While these waters undergo both isopycnal and diapycnal modification,
the distinct isopycnal component suggests that isopycnal mixing plays an important role in modulating the Fram Strait Branch
properties. Parts of the inflowing Fram Strait Branch waters are only slightly modified (cooled and freshened) to produce the
mixing convergence at $S > 35$ g kg$^{-1}$, $\Theta = 1.5-2$°C), which corresponds to the outflow on the same thermohaline properties
in Figure 4. We identify this as recirculating Atlantic Water, flowing westwards relatively quickly after entering the Arctic
Ocean. These Atlantic Waters are likely mixed with colder and fresher halocline waters or shelf waters near the shelf break
(Saloranta et al., 2004; Ivanov et al., 2024). In agreement with observations (Cottier and Venables, 2007), the curvature of the
transformation arrows in $T-S$ space towards higher densities additionally suggests the possibility of cabbeling as a relevant
transformation and densification process for the Fram Strait Atlantic Water.





Furthermore, Figure 5 shows the destruction through mixing of a cold and salty water mass ($S = 34.85 - 35.1$ g kg$^{-1}$, $\Theta < -0.5°$C). A regional decomposition (not shown) shows this water mass is seasonally produced during winter in the Barents Sea (see also Figure 11). Vectors indicate that these cold and salty Barents Sea waters mix with Atlantic Water originating

from Fram Strait. Being denser than the Fram Strait inflow, these waters likely contribute to the additional densification required for the transformation of Atlantic Water to the outflow density. Finally, mixing leads to a general convergence around the dense outflow waters ($\Theta = -1 - 0°$C, $S \approx 35.1$ g kg$^{-1}$) that ultimately exit through Fram Strait. This convergence requires contributions from both diapycnal and isopycnal mixing processes. Thus, surface forcing preconditions the inflowing Atlantic Water, generating large volumes of water near the outflow density, particularly from Barents Sea inflows. However, mixing ul-

timately sets the final $T$–$S$ properties of the dense outflow by homogenizing waters from the two inflow branches. Importantly, the relevant mixing processes are largely obscured by considering only the WMT in density space.

### 3.3 Overturning pathways and time scales

After analyzing the Arctic overturning in density and $T$–$S$ space, we next analyze the contributions of the different inflows at the gateways and the pathways of waters between the gateways. First, we decompose the total Eulerian overturning stream-

function into the net contribution from each of the four gateways (Figure 6a). Most of the lighter waters ($\sigma_0 < 27$ kg m$^{-3}$) enter the Arctic through Bering Strait and the Barents Sea, and exit the Arctic through Davis Strait and Fram Strait. More lighter waters exit Fram Strait in the hindcast than in the estimates from Tsubouchi et al. (2024). Most of the denser, Atlantic Waters (27.4 kg m$^{-3}$ < $\sigma_0$ < 27.8 kg m$^{-3}$) enter the Arctic through the Barents Sea and Fram Strait, and the Dense Waters ($\sigma_0$ > 27.95 kg m$^{-3}$) leave through Fram Strait. The net dense overturning across the Barents Sea Opening and Fram Strait is in

good agreement with the estimates from Tsubouchi et al. (2024), showing that the dense overturning and outflow of densified waters is dominated by Fram Strait.

The analysis in $T$–$S$ space indicated that both the Fram Strait and the Barents Sea branch contribute to the dense overturning across Fram Strait. To more accurately quantify the relative importance of the two branches, and along which pathways the waters are transformed, we use Lagrangian trajectories (Section 2.3), and explicitly track waters entering the four gateways. The

total Lagrangian streamfunction of all trajectories entering the Arctic (Fig. 6b) is very similar to the Eulerian streamfunction (Fig. 6a). Note that the Lagrangian streamfunctions are closed, since the trajectories conserve their volume transport along the way. Dense overturning is dominated by the Barents Sea Branch (e.g., Atlantic Water entering the Barents Sea and exiting Fram Strait), while the Fram Strait Branch has a major contribution at very high densities ($\sigma_0$ > 27.8 kg m$^{-3}$; Fig. 6b). This means that approximately 60% of Dense Water produced at the density of maximum overturning (27.95 kg m$^{-3}$) that flows

out of Fram Strait originates from the Barents Sea, and approximately 40% originates from Fram Strait itself. Furthermore, most of the water in the estuarine cell ($\sigma_0 < 27.5$ kg m$^{-3}$) originates from the Barents Sea and Davis Strait, which has mixed with fresher Arctic origin waters. Fram Strait waters also contribute to the estuarine cell, but at higher densities ($\sigma_0 > 27$ kg m$^{-3}$) and therefore mostly to the colder estuarine branch found in Figure 4. Finally, the Pacific overturning cell identified by Tsubouchi et al. (2024) and in Figure 4 emerges in the Lagrangian decomposition, but is hidden in the total overturning

streamfunction.





**Figure 6.** Eulerian and Lagrangian decomposition of Arctic overturning. a) Time mean Eulerian streamfunction across the Arctic gateways in ORCA0083-N06, and split into the net overturning across each gateway. b) Lagrangian overturning streamfunction for trajectories started at each gateway, until exiting the Arctic Ocean. Maps of trajectory probability for trajectories started at c) Fram Strait and d) the Barents Sea Opening.





The pathways of trajectories entering the Arctic at the two Atlantic Water gateways are shown in Fig.6c,d. Approximately 60% of the water entering Fram Strait recirculates and quickly flows back south. The rest largely follows the Atlantic Water Boundary Current (AWBC) eastwards along the rim of the Eurasian Basin (as indicated in Fig 1). Waters originating in the Barents Sea mostly enter the Arctic Ocean at St. Anna trough, where they merge with the AWBC. Further into the Arctic, north of the Laptev Sea, part of the Atlantic Water turns north and follows either the Lomonosov Ridge or the Gakkel Ridge, thereby recirculating in the Eurasian basin, whereas another part continues along the AWBC into the Amerasian Basin. These pathways are consistent with existing studies of the pathways of Atlantic Water in the Arctic (e.g., Timmermans and Marshall, 2020; Rudels and Carmack, 2022; Pasqualini et al., 2024).

Next, we briefly analyze the time scales of the Arctic overturning circulation. We do so by decomposing the Lagrangian overturning streamfunction based on how long the trajectories take to exit the Arctic (Fig. 7). Waters densifying and recirculating north of Fram Strait within less than 10 years contribute approximately 10% to the total maximum overturning. Waters recirculating in the Eurasian basin and following either the Lomonosov or Gakkel ridge back to Fram Strait take 10–20 years and contribute another 25% to the total overturning. This time scale is consistent with tracer studies of intermediate Atlantic Water circulation in the Eurasian basin (Wefing et al., 2021; Pasqualini et al., 2024). Approximately 65% of the total Arctic overturning is made up of water that takes longer than 20 years to exit the Arctic Ocean. In addition to trajectories circulating in the Eurasian basin for several decades, this also includes trajectories entering the Amerasian Basin and circumnavigating the entire deep Arctic basin. This indicates that most Atlantic Water will take multiple decades to exit Fram Strait as Dense Water and contribute to the lower limb of the AMOC.

The Arctic estuarine cell ($\sigma_0 = 27.0 \, \text{kg m}^{-3}$) has much shorter time scales, and most of the lighter waters exit the Arctic after 10–20 years. Here, the shortest pathway is the transpolar drift, a surface circulation feature that connects the Siberian Arctic to Fram Strait (not shown).

## 3.4 Geographical location of water mass transformation

The results in Section 3.1 show that the transformation of Atlantic Waters into Dense Waters is dominated by surface cooling (Figure 2). Mixing is driving the transformation into lighter polar waters and also plays a role in producing very dense overflow waters. Here, we remap the diapycnal water mass transformation from density space into geographical space to understand the importance of the two main pathways of Atlantic Water in the Arctic. First, we remap the surface-forced water mass transformation into geographical space. Second, we calculate where transformation occurs along the pathways of the water entering the Arctic Ocean via Fram Strait and the Barents Sea, using the Lagrangian trajectories. Third, to distinguish between the surface forced and the internal mixing component, we compare this total (Lagrangian) transformation with the (Eulerian) surface transformation, thereby combining the Lagrangian and Eulerian approaches.

We start with the surface forced water mass transformation (Fig. 8). The strongest transformation occurs in the Barents Sea and northwest of Svalbard, along the main inflow pathways of the Atlantic Water through Fram Strait and the Barents Sea Opening (Athanase et al., 2020; Skagseth et al., 2020). Most of the transformation in these areas occurs through surface cooling and through interaction with sea-ice melt and freeze. Negative surface transformation through sea-ice melt is largest



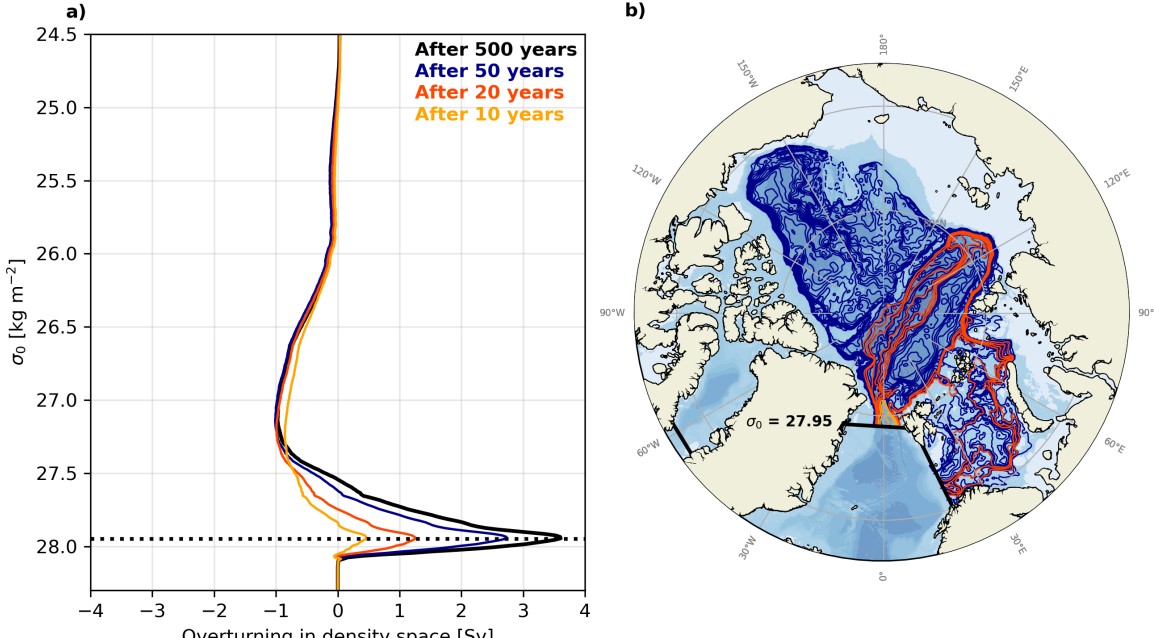

**Figure 7.** Timescales of the Arctic overturning circulation. a) Lagrangian Arctic overturning streamfunction for all trajectories leaving the Arctic after 10, 20, and 50 years, and over the entire 500 years of the Lagrangian experiment in ORCA0083-N06. b) Map showing the barotropic streamfunctions $\Psi$ of trajectories overturning at 27.95 kg m$^{-3}$ (i.e., enter the Arctic Ocean below this density and leave it above this density; dashed line in a). The different colors correspond to the time scales in a). $\Psi$ is set to zero over Greenland and the contour interval is 0.1 Sv.

north of Fram Strait. Further into the Arctic, the surface forcing is small, as those areas are permanently sea-ice covered and the warm Atlantic Waters are sheltered from the surface by the cold halocline (Aagaard et al., 1981).

Next, we look at the total water mass transformation in the Lagrangian experiment. For a given grid cell, we sum up the density change for all trajectories that pass through that grid cell, obtaining spatial maps of density divergence, or mass flux (Figure 9). For water entering via Fram Strait, a large part of the water mass transformation occurs close to Fram Strait (Fig.

9a). Waters flowing into the Arctic immediately northwest of Svalbard gain density due to surface cooling (Fig. 8), as these pathways are partly ice-free in winter (Athanase et al., 2020). Waters leaving the Arctic on the western side of Fram Strait also gain density on their way, while waters flowing north along the shelf break around the Yermak Plateau lose density. Because northwestern Fram Strait is usually ice-covered, this density change is likely through sea-ice melt (Fig. 8) and interaction between the inflowing Atlantic Waters and the outflowing Polar Waters or modified and recirculating Atlantic Waters. Mixing

of these water masses cools and freshens the inflowing Atlantic Waters, and warms and salinifies the outflowing waters (Fig. A2,A3). Overall, the salinity change dominates the density changes, such that Atlantic Water is becoming lighter and the outflowing waters become denser.





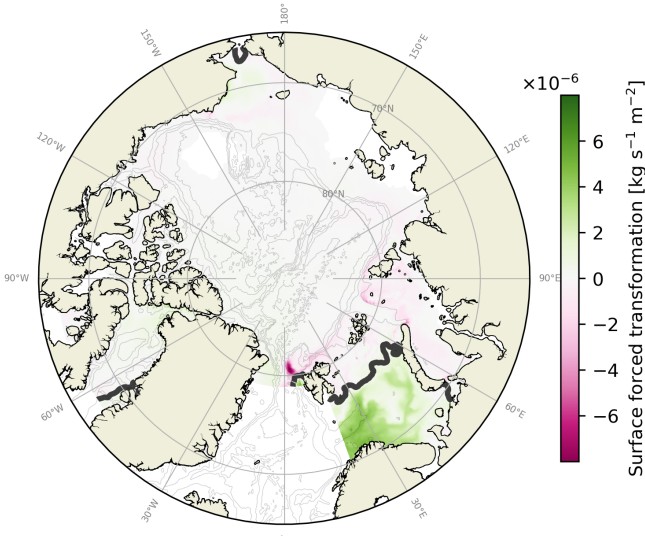

**Figure 8.** Locations of surface water mass transformation. Total surface-forced water mass transformation integrated over all density classes and averaged over 1979 - 2015 in ORCA0083-N06. The thick black line indicates the time-averaged sea ice extent (where there is more than 50% sea ice presence).

Further downstream, the Fram Strait branch waters slowly become less dense as they transit through the Arctic. Notable exceptions are found along the AWBC in the Nansen Basin, where waters gain density north of Franz Joseph Land and east
of St. Anna trough. Those locations coincide with troughs in the shelf break in the northern Barents and Kara seas, where the Fram Strait branch interacts with waters originating in the Barents Sea (Schauer et al., 2002; Ivanov et al., 2024), and where the mass divergence shows a density loss for Barents branch waters (Fig. 9b). This indicates that the interaction between the two Atlantic Water branches transforms some of the Fram Strait branch waters into denser waters, and contributes to the dense water formation along the Fram Strait pathway, consistent with results in Section 3.2. Analysis of the heat and salinity change
shows that the heat exchange between the two branches dominates the density signal in these regions (Fig. 5b, A2,A3).

Waters entering the Barents Sea undergo a strong density increase in the Barents Sea as they are cooled (Fig. 9b), consistent with Figure 8. Beyond the Barents Sea, waters decrease their density along the Siberian shelves, most strongly close to the outflows of Siberian rivers, where they mix with the fresh riverine water, decreasing their salinity (Fig. A3). Water that becomes lighter will mostly follow the transpolar drift and eventually exit the Arctic with a lower density, contributing to the estuarine
cell. Before exiting the Arctic, just north of Fram Strait, those waters increase their density again, most likely by mixing with the inflowing and recirculating Atlantic Waters in the central Fram Strait, which, as shown above, experience a decrease in density (Fig. 9a).





Lastly, waters from both branches decrease their density in the boundary current north of Alaska (Fig. 9a,b). Here, they mostly interact with fresher Pacific Waters originating from Bering Strait that enter the Amerasian Basin from the Chukchi

Shelf, or that circulate in the Beaufort Gyre, and ventilate the Arctic halocline (Aagaard et al., 1981).

Next, we focus on the water mass transformation that contributes to the dense overturning circulation by only selecting trajectories that contribute to net overturning at $\sigma_0$ = 27.95 kg m$^{-3}$ (the density of maximum overturning; Fig. 9c,d). Apart from the Barents Sea and the areas north of Svalbard, most locations of transformation for both branches are now over the Atlantic Water Boundary Current (AWBC). Waters originating from Fram Strait increase their density along most of their path

along the AWBC, and notably north of Franz Joseph Land and St. Anna trough, where they, as discussed, likely interact with Barents Sea waters. Waters originating from the Barents Sea increase in density within the Barents Sea, and decrease in density downstream where they interact with the Fram Strait branch.

Further downstream in the Laptev Sea, Barents Sea origin waters in the boundary current increase their density (Fig. 9d). The Laptev shelf is a known area of high sea-ice production and formation of dense shelf waters (Cornish et al., 2022), which

could mix with the Atlantic Waters in the AWBC and increase their density. An analysis of dense water production over the Laptev shelf in ORCA0083-N06 confirms that there are dense waters produced, although slightly lighter than the density of maximum Arctic overturning (not shown). Another source for the density increase in this region is Barents and Kara Sea origin waters that flow through Vilkitsky Strait, gain density on the shelf, and then mix with the waters in the AWBC (Figure 9d; Janout et al. (2017)).

As a last step, we combine the surface forced transformation in Fig. 8 and the total Lagrangian transformation in Fig. 9c,d to produce a rough estimate of the relative contributions of surface forcing and internal mixing in driving water mass transformation at the density of maximum overturning. We do so by determining the regions with substantial surface forcing (more than $+1 \times 10^{-6}$ kg s$^{-1}$ m$^{-2}$; Fig. 8) . The areas meeting this criteria, e.g., the Barents Sea and the region northwest of Svalbard, are indicated in Fig. 9c,d (hatched areas). Assuming that water in those regions will predominantly transform through

surface forcing, we sum the transformation from Figure 9c,d inside the regions and compare them to the transformation outside the regions, which we then assume to be due to internal mixing. Based on this calculation, we estimate that for the dense overturning cell, approximately 15 % of total transformation for the Fram Strait branch is surface-forced, and approximately 85 % driven by internal mixing. For the Barents Sea branch, approximately 85 % of the transformation is surface-driven, and only approximately 15% is through internal mixing. These rough estimates are robust to the exact choice of threshold used

to define regions of surface forcing. Taking the two branches together gives an estimated total contribution of approximately 25% from internal mixing to the dense overturning, which is consistent the earlier result that surface forcing dominates the transformation, but internal mixing plays a role at high densities (Fig. 2, 5).

## 4 Discussion and Conclusions

The Arctic is the northernmost terminus of the AMOC, and produces some of the densest waters that enter the AMOC lower

limb (Zhang and Thomas, 2021). Despite its importance, the Arctic overturning circulation remains little explored. Although





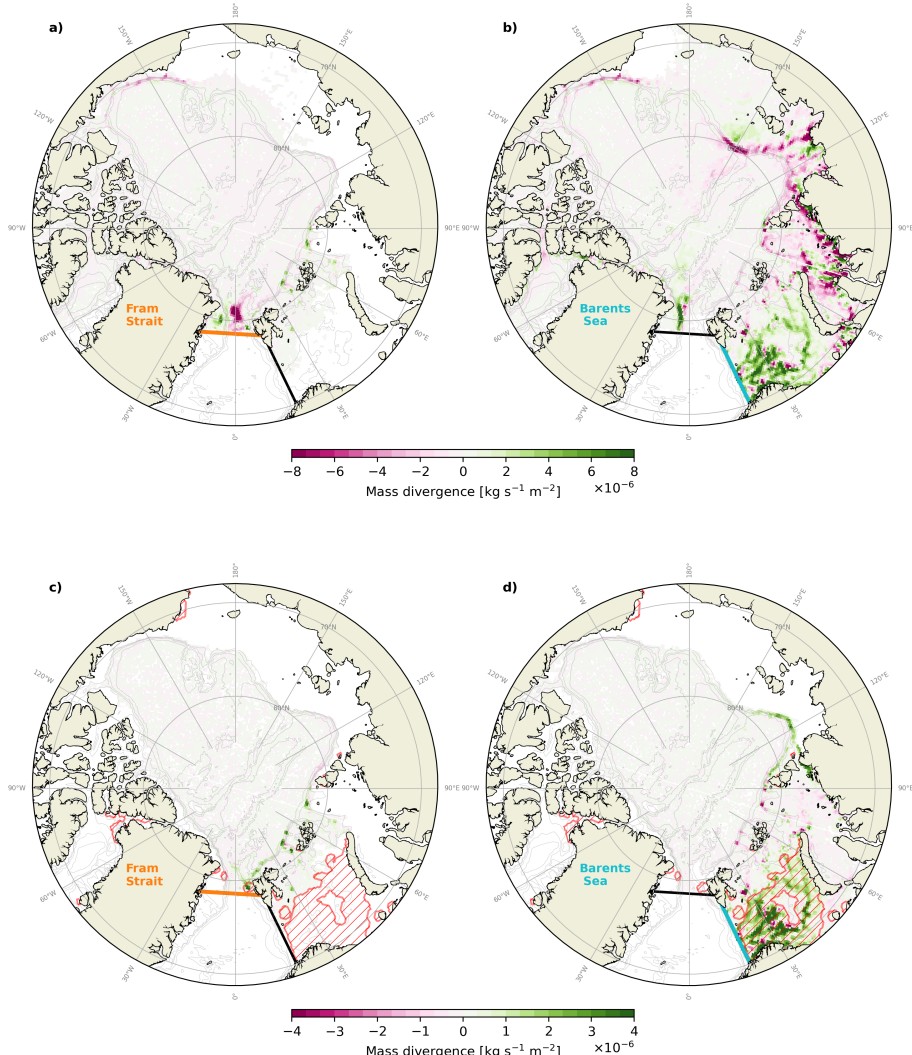

**Figure 9.** Locations of water mass transformation. Total mass divergence of a,b) all trajectories entering the Arctic through Fram Strait and the Barents Sea and c,d) for trajectories overturning at 27.95 kg m$^{-3}$ (i.e., entering the Arctic below this density, and leaving it above this density). Values are normalized by the grid area. Green colors indicate mass (density) gain, purple colors indicate mass loss. Hatched red area indicate regions where surface-forced transformation in Figure 8 is larger than $1 \times 10^{-6}$ kg s$^{-1}$ m$^{-2}$, which are used to calculate the contribution of surface forcing to water mass transformation.

it is well established that the Arctic Ocean transforms the inflowing warm, salty Atlantic Water into both cold, dense waters, and cold, fresh surface waters (Eldevik and Nilsen, 2013; Rudels et al., 2015; Brown, 2019; Haine, 2021), how much transformation occurs and where and by which mechanisms this takes place is still not established. Here, we have quantified the mean



structure of the Arctic overturning circulation in the eddy-rich ORCA0083-N06 ocean hindcast between 1979 and 2015 using
an Eulerian water mass transformation framework and Lagrangian experiments. Key results on the transformation mechanisms
and locations are summarized in Figure 10.

We find that Atlantic Water is transformed into dense waters at rate of 3.1 Sv and into Polar Water at a rate of 1 Sv, consistent
with recent estimates based on reanalysis and observational estimates (Tsubouchi et al., 2024; Årthun et al., 2025). The dense
overturning is dominated by the Barents Sea branch, which transforms Atlantic Water over a wide range of densities into
dense waters, while the Fram Strait branch contributes strongly only to a narrow density range around the density of maximum
overturning ($\sigma_0$ = 27.95 kg m$^{-3}$). The dominance of the Barents Sea in the dense water production in the Arctic Ocean
confirms earlier work (Smedsrud et al., 2013; Moat et al., 2014). Atlantic Water is transformed into dense waters mostly by
surface cooling in the Barents Sea and north of Svalbard (Fig. 5a, 8, 9). Additionally, interior mixing plays an important role
in transforming Atlantic Water into dense waters through two main mechanisms: First, by mixing cold, dense Barents branch
waters with warmer and more saline Fram Strait waters on the shelf break in the Nansen Basin, and in St. Anna Trough at the
Barents Sea exit (Fig. 5b, 9c,d). Secondly, Atlantic Waters increase in density by mixing with dense shelf waters in the Kara
and Laptev seas. We estimate the contribution of interior mixing to dense overturning to be approximately 25% in total, with
a much higher contribution to the Fram Strait branch and a small contribution to the Barents Sea branch. In contrast, mixing
dominates the estuarine branch by mixing Atlantic Waters with fresher waters. This mixing mostly occurs immediately north
of Fram Strait, and along the Siberian coast, where the largest rivers enter the Arctic Ocean. These results are consistent with
recent estimates based on inverse modeling, which find a large role of interior mixing for the estuarine branch, but a smaller
role for the dense overturning (Brown, 2019; Bacon et al., 2022).

An analysis of the time scales involved in the Arctic overturning circulation shows that most of the dense waters produced
along the Fram and Barents Sea branches take multiple decades to exit Fram Strait. This opens the question of how quickly
changes in water mass transformation will translate into changes in the overturning strength. Recently, Årthun et al. (2025)
found a strengthening of the dense Arctic overturning between 1993 and 2020. This strengthening corresponds to sea-ice loss
and increased surface transformation in the Barents Sea and north of Svalbard, but, given the long overturning timescales we
find, it is surprising to see such an immediate response in overturning to changes in surface forcing. A more detailed analysis
of recent changes in the Arctic overturning circulation that reconciles these findings will be the topic of another study.

In this study, we have focused on annually averaged water mass transformations and overturning. There is additionally
large seasonality in the mechanisms of overturning, especially in the surface forcing (Fig. 11). In winter, surface forcing
produces both cooled and densified Atlantic Water from the Barents Sea and dense, brine-enriched waters near the freezing
point from sea ice formation. These are subsequently homogenized through mixing into dense shelf waters. Approximately
opposite transformations occur in the summer months. In the annual average, however, these dense shelf waters are essential
for densifying Fram Strait Branch waters to outflow densities (Figure 5b). Notably, the Barents Sea Branch experiences stronger
freshening from mixing during winter than summer, despite the greater availability of freshwater in summer. This is likely due
to weaker stratification in winter, which allows wind-driven mixing and cooling-induced deepening of the mixed layer to mix
freshwater down into the Atlantic Water layer. The convergence of outflow waters, both recirculated Fram Strait Atlantic Water



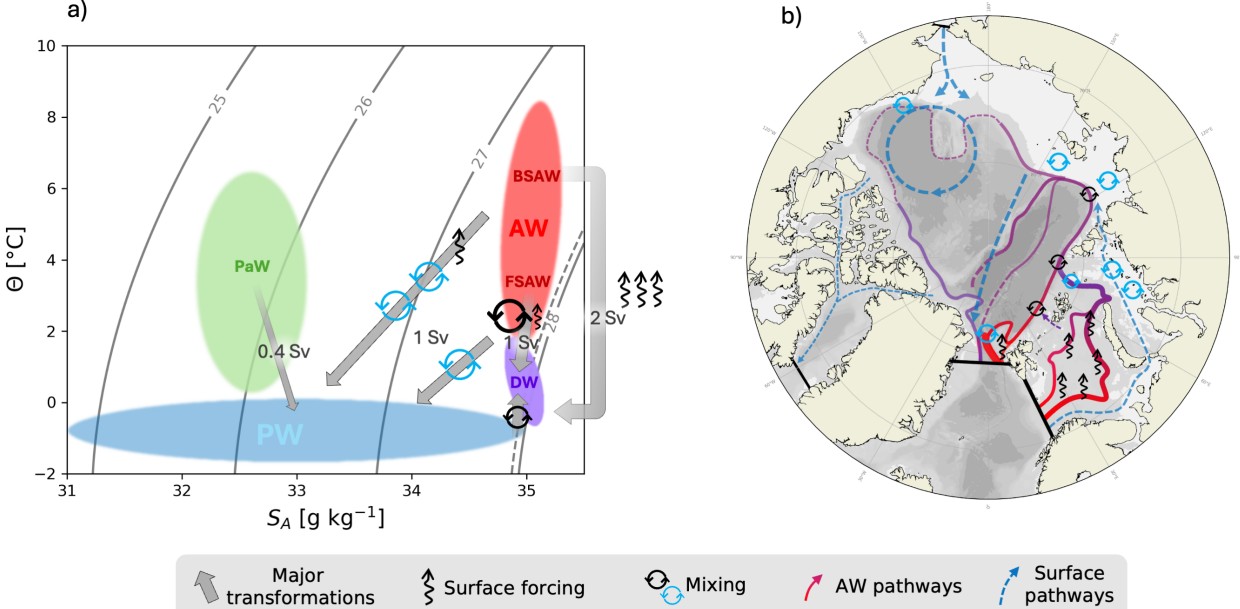

**Figure 10.** Schematic of the pathways and mechanisms associated with the Arctic overturning circulation. a) Temperature-salinity diagram showing the major transformations of Atlantic Water (AW) into dense waters (DW) and Polar Waters (PW) and Pacific Water (PaW) into PW. The AW is further split into the Barents Sea (BSAW) and the Fram Strait (FSAW) components. Numbers in Sv indicate the approximate rate of transformations. b) map indicating the pathways and locations of transformation, showing that AW is modified by surface forcing (heat loss and ice melt/freeze) in the Barents Sea and north of Svalbard, and further modified by interior mixing mainly with denser water along the boundary current in the Nansen Basin (black circular arrows) and with lighter waters along on the Siberian shelves and north of Fram Strait (blue circular arrows).

and the densest components of the outflow (Fig. 4,5b), show minimal seasonal variability. Finally, the estuarine cell's seasonal
variability is set by freshwater mixing along the Siberian Coast, which is stronger in summer.

Our results are based on a single eddy-rich ocean hindcast and there are inherent uncertainties in our quantification of pathways, timescales and the contribution of mixing. The model does not include tides and is not eddy-resolving on the shelves, and might thus underestimate mixing (Rippeth et al., 2015; Fer et al., 2015; Janout and Lenn, 2014; Renner et al., 2018). However, even though the hindcast simulation has biases related to the Atlantic Water layer temperature and fresh
outflows at the gateways (Fig. 6,A1), it is able to realistically simulate key components of the Arctic Ocean circulation (Kelly et al., 2018, 2019), including Atlantic Water inflows, pathways, timescales and strength of dense overturning.

Our study presents, for the first time, a comprehensive analysis of the hydrographic and spatial structure of the Arctic overturning circulation based on the period 1979–2015. During this period the Arctic Ocean experienced large changes in e.g., sea ice extent and hydrography (Stroeve and Notz, 2018; Polyakov et al., 2017), and further warming and sea ice loss are



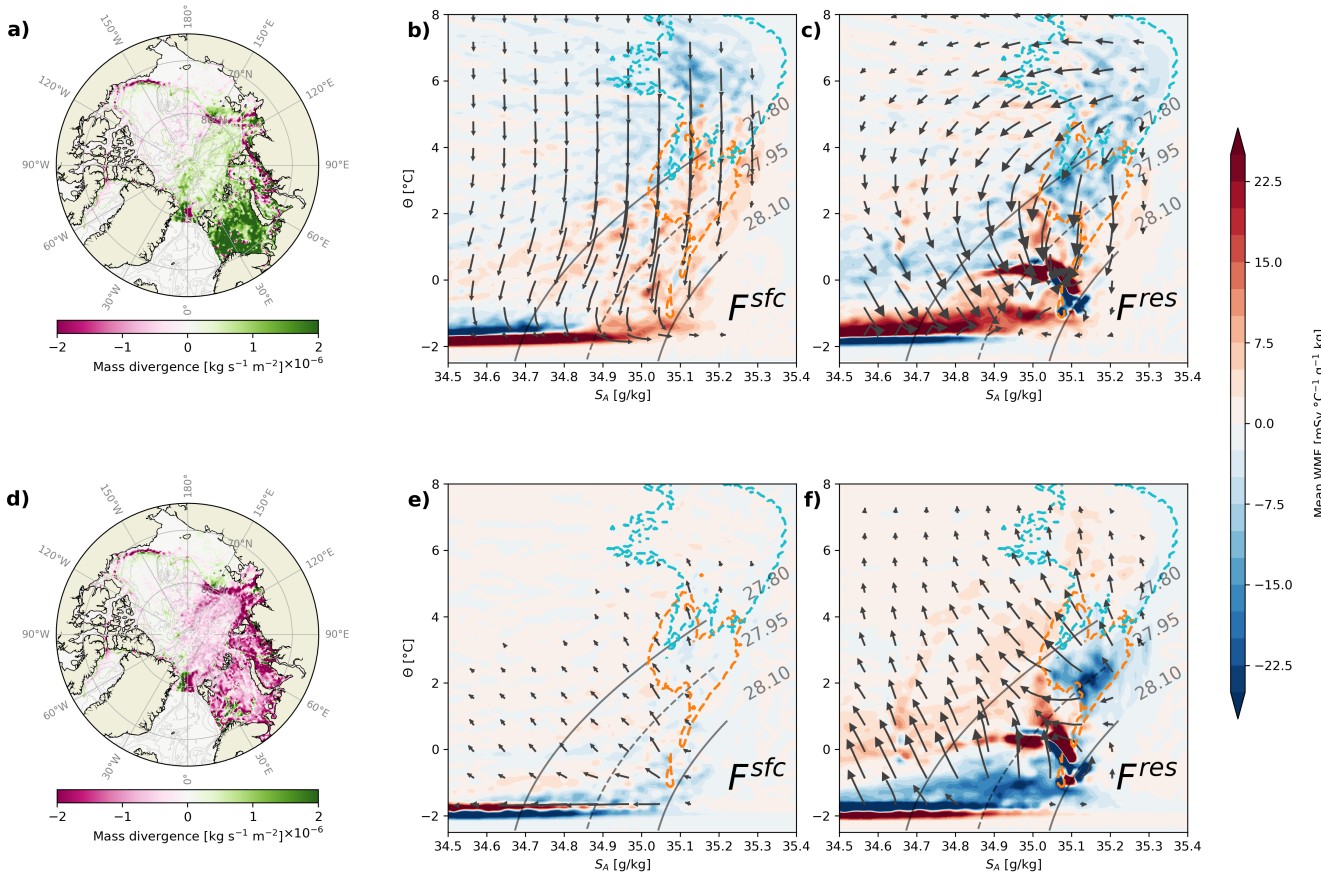

**Figure 11.** Seasonality of Arctic water mass transformation. Total water mass transformation during winter (October - March, a-c) and summer (April - September, d-f) from Lagrangian (a,d) and Eulerian (b,c,e,f) perspectives. a,d) as Fig. 9a,b, but for all trajectories started at Fram Strait and the Barents Sea Opening, and for winter and summer, respectively. b,c) as Fig. 5, but for winter. e,f) as Fig. 5, but for summer.

expected (Årthun et al., 2019; Notz and SIMIP community, 2020; Dörr et al., 2024). Our results can thus be used as a baseline when analyzing and interpreting recent and future changes in the Arctic overturning circulation and their causes and impacts.



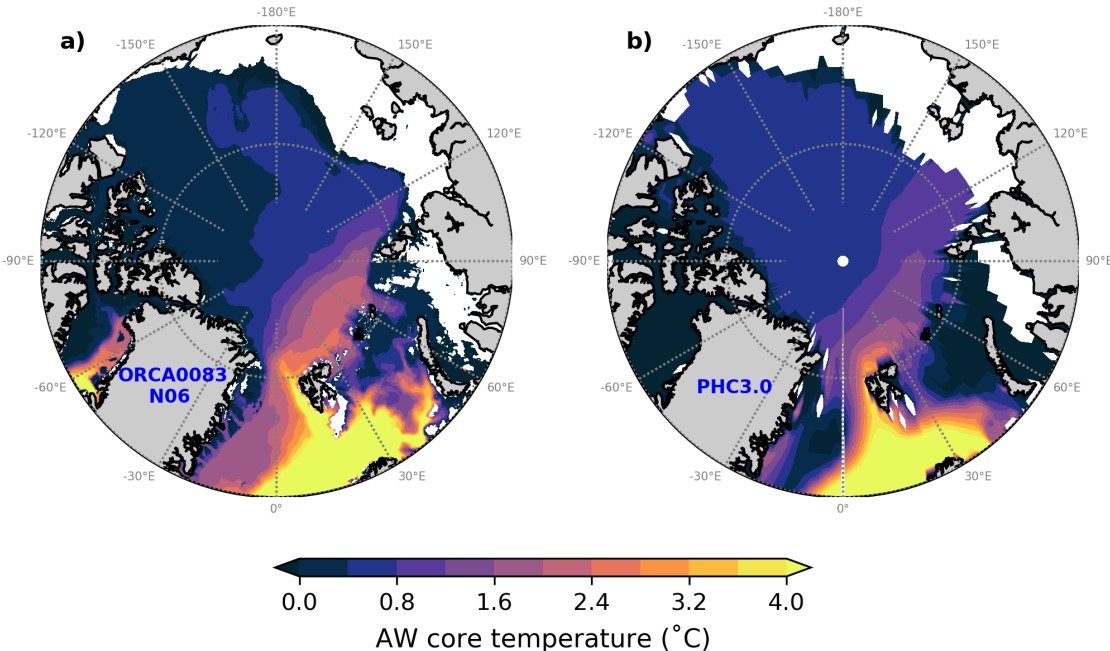

**Figure A1.** Comparison of the Atlantic Water core temperature in a) ORCA0083-N06 and b) PHC3.0 (Steele et al., 2001). The core temperature is the maximum temperature of the water column where the salinity is above 34.7. Shelf regions with a depth lower than 100 m are not shown.



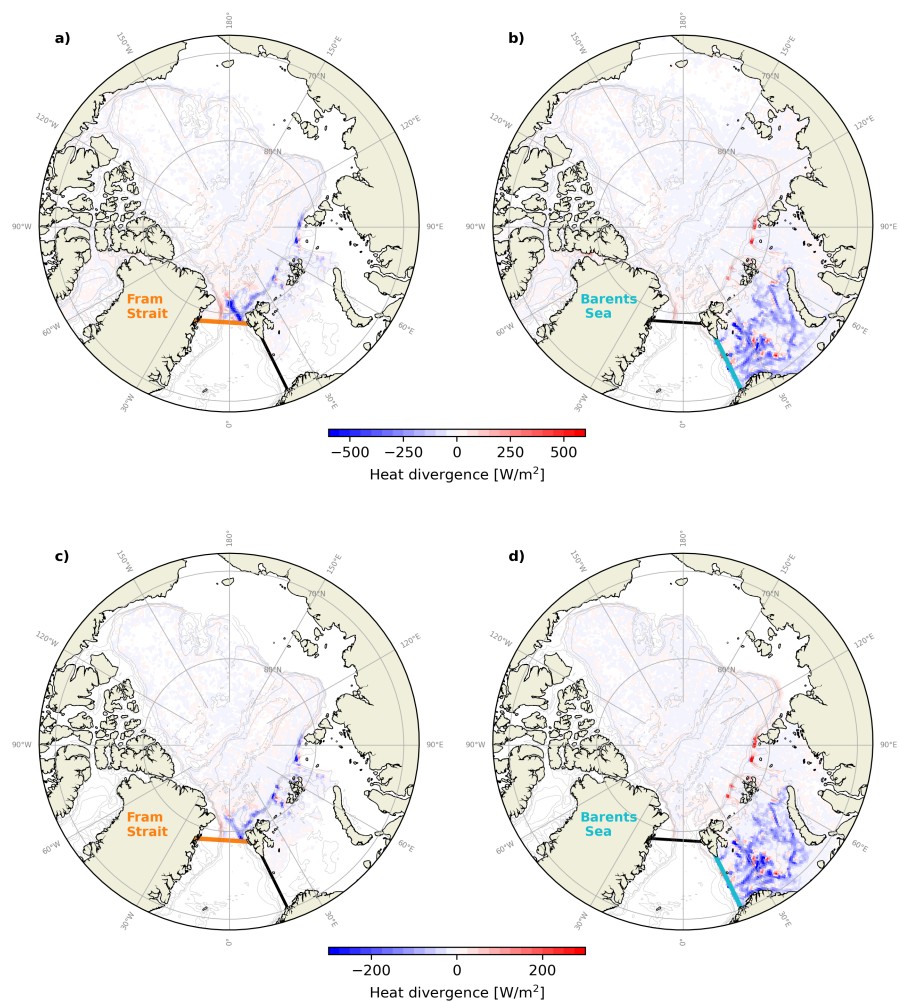

**Figure A2.** As Fig. 9, but for heat divergence.



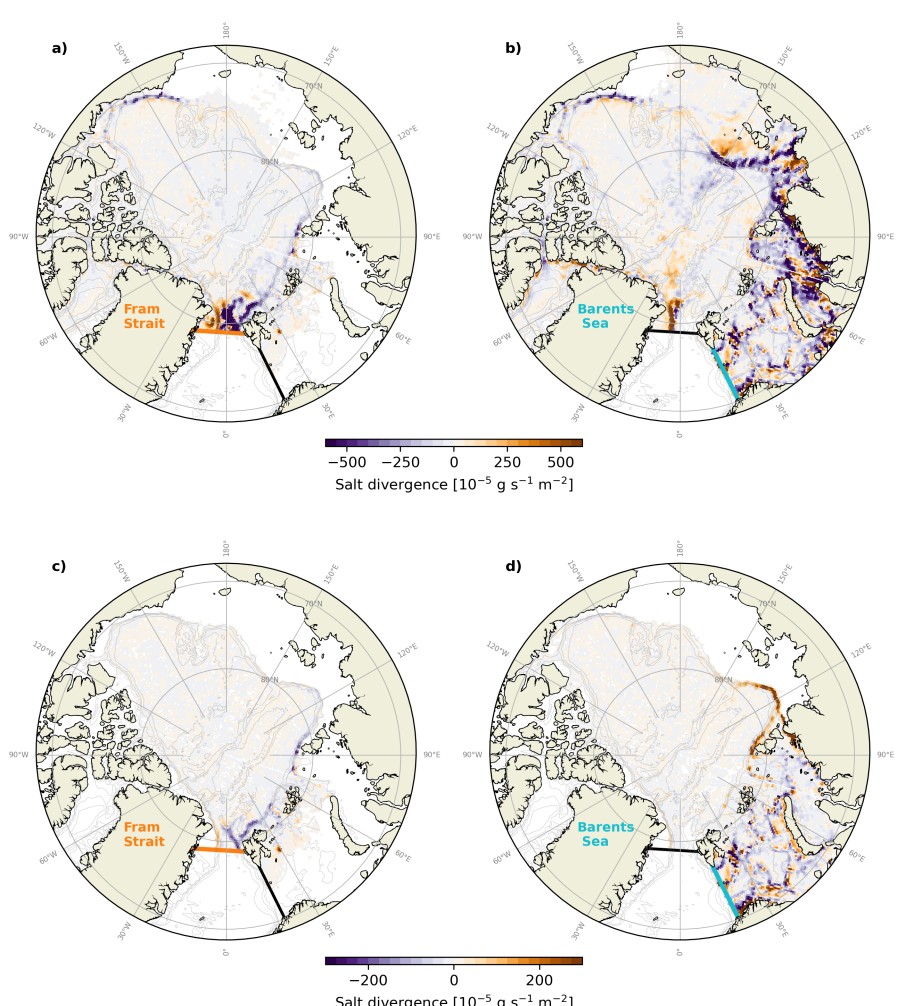

**Figure A3.** As Fig. 9, but for salt divergence.



*Code and data availability.* Monthly output from the ORCA0083-N06 hindcast can be found at https://gws-access.jasmin.ac.uk/public/ nemo/runs/ORCA0083-N06/means/. The source code of TRACMASS version 7.1 is available at https://github.com/TRACMASS/Tracmass/ releases/tag/v2021.10%2Fv7.1, updated from Aldama-Campino et al. (2020). The code for the Water Mass Transformation analysis is available from https://github.com/dgwynevans/wmt. The PHC3.0 climatology is available from https://odv.awi.de/data/ocean/phc-30/. Lagrangian trajectory data produced in this study can be found in Dörr (2025).

*Author contributions.* JD, CM and MÅ conceived the study. MÅ acquired the funding for this study. JD and CM carried out all analysis and prepared the original draft. All authors interpreted the results, and reviewed and edited the final version of the manuscript.

*Competing interests.* The contact authors have declared that none of the authors has any competing interests.

*Acknowledgements.* This work was funded by the Research Council of Norway project Overturning in the new Arctic (Grant 335255).



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
