# Peer review of "The Arctic overturning circulation: transformations, pathways and timescales"

_EGUsphere, 2025_

## Referee Comment (RC1)

**Review of egusphere-2025-4345**
**Dörr et al., 2025**
**Ocean Science**

**General comments**

In this study, the authors analyze the overturning circulation and water mass transformation of the Arctic Ocean in a 1/12° global ocean model hindcast simulation over 1979–2015. The analysis of water mass transformation is performed both in density space and temperature–salinity space. In addition, pathways and timescales of the circulation are determined using Lagrangian virtual particle tracking. These approaches allow to quantify the contribution of surface forcing and mixing in different regions to the full water mass transformation. Results include the identification of the Barents sea as a major location of surface-forced dense water formation, and the time scales of different circulation routes in and out of the Arctic.

In my opinion, this paper provides an excellent and detailed overview of the circulation and water mass budgets of the Arctic (in this particular ocean model). The analysis is extensive and thorough, and the text and figures are of high quality and clarity.

I recommend that this paper be accepted for publication after addressing the minor comments below.

**Specific comments**

**Literature**  The results of the WMT analysis in T-S space in this paper should be compared to the following study which employed a similar approach: Pemberton, P., J. Nilsson, M. Hieronymus, and H. E. M. Meier, 2015: Arctic Ocean Water Mass Transformation in S–T Coordinates. J. Phys. Oceanogr., 45, 1025–1050, https://doi.org/10.1175/JPO-D-14-0197.1.

**Ln. 64**  How does the use of monthly mean velocities (instead of a higher time resolution) affect the applicability of the Lagrangian tracking algorithm? Did the setup include some stochastic element to account for unresolved turbulence? This should either be detailed in the Methods or commented on in the Discussion.

**Section 2.2**  Since the many equations in this section make it look a bit "dense", it could be useful to separate it into two subsections (e.g., 2.2.1 "Density space" and 2.2.2 "T–S space"). I leave this choice to the authors of cours of coursee.

**Ln. 80**  Since you mention "previous studies", it would be good to explicitly cite them. Currently it is not clear whether these previous studies also applied the Walin framework to the Arctic ocean, or if they simply also used the Walin framework in some other way.

**Technical corrections**

**Figure sizes**  In some of the figures, the text labels appear smaller than in others. In particular, the longitude labels on polar projection plots are so small as to be illegible in most cases. I recommend checking the consistency of font sizes across all figures.

**General**  In all the mathematical symbols with "sfc" and "res" superscripts, you should probably use `mathrm` to avoid the superscripts looking like $s \cdot f \cdot c$. Example: $F_\Theta^{\mathrm{sfc}}$ vs. $F_\Theta^{sfc}$

**Ln. 11** although "northern" overturning is clearly correct, perhaps writing "Atlantic" over-turning would make the broader impact of this paper clearer (AMOC slowdown, etc.)

**Ln. 86** Maybe explicitly state *volume* transport

**Eq. 3** Are the units in this equation consistent? It seems that the last two terms currently have different units given their different denominator/differentiation variable.

**Ln. 101** Use $v$ instead of $\nu$ in the equation

**Fig. 1** Consider adding a vertical line at $x = 0$. Also, the $y$-axis label $\sigma_0$ appears to be smaller than the rest of the text.

**Ln. 191** The salinity range should not be in parentheses

**Fig. 3** In the colorbar label, the salinity units should probably be inverted

**Ln. 208** undergo $\rightarrow$ undergoes

**Ln. 231** superfluous closing parenthesis

**Fig. 8 caption** Why is 50% sea ice concentration used as a threshold for sea ice extent (instead of the more common 15%)? In any case, since this is mostly for illustrative purposes in this figure, this is probably not important.

**Ln. 364** "the AMOC lower branch" $\rightarrow$ the AMOC's lower branch / the lower branch of the AMOC

**Fig. 9 caption** grid *cell* area

**Ln. 372** at *a* rate

---

## Author Comment (AC1)

**Response to Reviewer #1**

**Note: In this response, the reviewers comments have been left untouched, and we added our responses in bold.**

General comments
In this study, the authors analyze the overturning circulation and water mass transformation of the Arctic Ocean in a 1/12° global ocean model hindcast simulation over 1979–2015. The analysis of water mass transformation is performed both in density space and temperature– salinity space. In addition, pathways and timescales of the circulation are determined using Lagrangian virtual particle tracking. These approaches allow to quantify the contribution of surface forcing and mixing in different regions to the full water mass transformation. Results include the identification of the Barents sea as a major location of surface-forced dense water formation, and the time scales of different circulation routes in and out of the Arctic. In my opinion, this paper provides an excellent and detailed overview of the circulation and water mass budgets of the Arctic (in this particular ocean model). The analysis is extensive and thorough, and the text and figures are of high quality and clarity.

I recommend that this paper be accepted for publication after addressing the minor comments below.

**We thank the reviewer for the positive and constructive comments. We respond to the comments below.**

Specific comments

Literature
The results of the WMT analysis in T-S space in this paper should be compared to the following study which employed a similar approach: Pemberton, P., J. Nilsson, M. Hieronymus, and H. E. M. Meier, 2015: Arctic Ocean Water Mass Transformation in S–T Coordinates. J. Phys. Oceanogr., 45, 1025–1050, https://doi.org/10.1175/JPO-D-14-0197.1.

**We thank the reviewer for pointing us to relevant literature. In the revised manuscript we have expanded the discussion on how our results compare to previous estimates of water mass transformation in the Arctic Ocean, including the paper by Pemberton et al. (2015) (L. 393-394, L.426-430, L. 407-417).**

Ln. 64 How does the use of monthly mean velocities (instead of a higher time resolution) affect the applicability of the Lagrangian tracking algorithm? Did the setup include some stochastic element to account for unresolved turbulence? This should either be detailed in the Methods or commented on in the Discussion.

**We have tested using 5-day time-evolving fields as input, and looping over the period 1979 - 2015 instead, but the main results did not change. We do not add any stochastic diffusion as this would break volume conservation (which is essential for our calculations of e.g., the Lagrangian streamfunctions), and instead follow water using the resolved advective pathways in the model. We now mention this in the methods in L. 137-138, L. 144-145**

Section 2.2 Since the many equations in this section make it look a bit "dense", it could be useful to separate it into two subsections (e.g., 2.2.1 "Density space" and 2.2.2 "T–S space"). I leave this choice to the authors

of course.

**We have followed the reviewer's suggestion and divided Section 2.2 into 2.2.1 and 2.2.2.**

Ln. 80 Since you mention "previous studies", it would be good to explicitly cite them. Currently it is not clear whether these previous studies also applied the Walin framework to the Arctic ocean, or if they simply also used the Walin framework in some other way.

**Good point. We have added a few studies who have used the Walin (water mass transformation) framework for the Arctic (L. 82).**

Technical corrections Figure sizes In some of the figures, the text labels appear smaller than in others. In particular, the longitude labels on polar projection plots are so small as to be illegible in most cases. I recommend checking the consistency of font sizes across all figures.

**We have adjusted the labels/fontsizes in Figure 6, 7, 8, 9, 11, and A3, and A4.**

General In all the mathematical symbols with "sfc" and "res" superscripts, you should probably use mathrm to avoid the superscripts looking like s f c. Example: $F_\Theta^{\mathrm{sfc}}$ vs $F_\Theta^{sfc}$

**This has now been adjusted.**

Ln. 11 although "northern" overturning is clearly correct, perhaps writing "Atlantic" over- turning would make the broader impact of this paper clearer (AMOC slowdown, etc.)

**We have replaced 'northern' by 'Atlantic' in L.12 as suggested.**

Ln. 86 Maybe explicitly state volume transport

**Done.**

Eq. 3 Are the units in this equation consistent? It seems that the last two terms currently have different units given their different denominator/differentiation variable.

**The formula has been corrected.**

Ln. 101 Use $v$ instead of $\nu$ in the equation

**Done.**

Fig. 1 Consider adding a vertical line at x = 0. Also, the y-axis label $\sigma_0$ appears to be smaller than the rest of the text

**The size of the y-axis label in Figure 2 has been fixed. We have not added a vertical line at x = 0 as the figure is already quite busy.**

Ln. 191 The salinity range should not be in parentheses

**We have removed the parentheses.**

Fig. 3 In the colorbar label, the salinity units should probably be inverted

**We have switched the order of the units.**

Ln. 208 undergo → undergoes

**Done.**

Ln. 231 superfluous closing parenthesis

**We have removed the parenthesis.**

Fig. 8 caption Why is 50% sea ice concentration used as a threshold for sea ice extent (instead of the more common 15%)? In any case, since this is mostly for illustrative purposes in this figure, this is probably not important.

**We have changed 'sea ice extent' to 'sea ice edge (defined as 50% sea-ice cover)' in the caption. This is not an uncommon definition, and is here indeed only shown for illustrative purposes.**

Ln. 364 "the AMOC lower branch" → the AMOC's lower branch / the lower branch of the AMOC

**We have changed the text to 'the lower branch of the AMOC' (L. 384).**

Fig. 9 caption grid cell area

**Done.**

Ln. 372 at a rate

**Done.**

**References**

Pemberton, P., Nilsson, J., Hieronymus, M., and Meier, H. E. M. (2015). Arctic Ocean Water Mass Transformation in S–T Coordinates. *Journal of Physical Oceanography*, 45(4):1025–1050.

---

## Author Comment (AC2)

**Response to Reviewer #2**

**Note: In this response, the reviewers comments have been left untouched, and we added our responses in bold.**

Overall comments:

This is an interesting and useful paper on Arctic Ocean overturning circulation. I agree with the authors that "it is important to understand how Atlantic Water entering the Arctic Ocean is transformed" (line 46) and that the "the relative contributions of surface forcing and interior mixing are not" known (line 48), or "where along the Atlantic Water pathways the transformation is most pronounced". These are all good topics to study that are relevant to the readers of Ocean Science. The methods are appropriate, well-described, and well-executed and the results are well supported. The conclusions are useful for oceanographers studying the Arctic overturning circulation and water mass transformation.

The paper is clear, well-organized, accurate, and well-written with an appropriate title and abstract. With a few important caveats (see below), the paper is novel and well situated in the literature on Arctic Ocean overturning.

In terms of scientific significance, the paper rates as Good.

In terms of scientific quality, the paper rates as Good.

In terms of presentation quality, the paper rates as Excellent.

I have three related major suggestions on connecting the present paper with prior studies, and several minor suggestions. I recommend the paper is returned to the authors for a major revision then reconsidered for publication by Ocean Science.

**We thank the reviewer for their thorough and constructive comments. We respond to the specific comments below.**

Specific major comments:

The present paper should carefully consider a closely related paper on Arctic Ocean water mass transformation by Pemberton et al. (2015, JPO, 10.1175/JPO-D-14-0197.1, it's not cited in the present paper). It's important to compare and contrast the present results with those in this earlier paper. For instance, Figures 4, 5, and 11 of the present paper show the same quantities as various Figures in Pemberton's paper. Of course, there are important differences between the two studies, like the refined model resolution and spatial information in the present paper. A careful discussion to compare and contrast the two studies is needed. For instance, Pemberton et al. discuss the importance of their surface salinity restoring on their estimates of water mass transformation (it's not negligible, e.g., see their conclusion). The present model also includes surface salinity restoring, and fixing this unrealistic aspect of the model configuration is an important next step.

Also, the present paper should discuss the results of Tsubouchi et al. (2024, which is cited in the present paper) in more depth. For instance, Figures 4 and 6 of the present paper show the same quantities as Figure 4 of Tsubouchi et al. (2024). Again, there are important methodological differences, but a careful

discussion is needed. For instance, the overturning in density space shown in Figure 4 of the present paper has a significantly different split between the Barents Sea and Fram Strait compared to Tsubouchi's paper (see line 252). Because Tsubouchi et al.'s estimates are (mainly) based on observations at gateway sections, they don't suffer from the surface-salinity-restoring issue mentioned above (although they have other issues, of course). Some discussion of the possible reasons for the differences, and therefore, the pros and cons of each study, would be helpful.

Finally, the present paper should discuss the new study by Brown et al. (2025, AGU Advances, 10.1029/2024AV001529, it's not cited in the present paper). Brown et al. update and extend the Tsubouchi et al. (2024) paper. Brown et al.'s results represent the current best estimate of Arctic water mass transformation from gateway observations and surface flux reanalyses. The present paper should compare and contrast its results with their study, and carefully discuss the reasons for the differences and the pros and cons of each approach.

**We thank the reviewer for pointing us to relevant literature. In the revised manuscript we have expanded the discussion on how our results compare to previous estimates of water mass transformation in the Arctic Ocean, including the papers by Pemberton et al. (2015), Tsubouchi et al. (2024), and Brown et al. (2025), highlighting the different approaches used and their strengths and weaknesses (L. 393-394, L.426-430, L. 407-417).**

Specific minor comments:

Abstract: The last sentence mentions how this paper "contributes to understanding...future changes" in the Arctic overturning circulation. This is mentioned again in the final sentence of the main text (lines 415–416), where it talks about establishing a baseline of Arctic overturning. This is all fine, but the abstract had me expecting something more involved, so I suggest you mention "baseline" in the final sentence of the abstract too.

**As suggested by the reviewer, we now mention 'baseline' in the last sentence of the abstract (L. 10 - 12)**

Line 73: Describing Beszczynska-Moller et al's 2012 paper as "recent" stretches the definition of "recent" a bit.

**We have removed 'recent' from the sentence (L. 74).**

Line 95: The final two terms seem inconsistent with equation 5. Should they be the derivatives of $G\_\Theta$ and $G\_S$ (not $G\_{S \Theta}$? If not, how is $G\_{S \Theta}$ connected to $G\_\Theta$ and $G\_S$?

**We have corrected the erroneous formulation pointed out by the reviewer.**

Line 120: It talks about the residuals including "sea-surface restoring". Remind the reader here that this model has surface salinity restoring (line 63) that will appear in the residual term.

**We now specify that the sea surface salinity restoring is included in the surface freshwater flux (L. 115-117).**

Line 155: Is the "long term trend of buoyancy gain in the Arctic Ocean deep waters" a real physical signal? Or is it model drift? (Or something else?).

**As the focus of this study is the mean state of the overturning circulation we have not investigated this long term trend. We note though that the long term buoyancy gain is consistent with observed warming of Arctic dense waters. This is now specified in the text (L.165-166).**

Line 215: It talks about the formation of the densest waters in the Arctic (saline, freezing Barents Sea water). State the density, salinity and temperature of these waters in the model.

**We have added the appropriate salinity and density of the Barents Sea dense waters in the model in L. 232-233**

Lines 264–265: Explain how the 60% and 40% numbers are found.

**We have rephrased this to (in L. 283-285.) 'Comparing the values of the streamfunctions at the density of maximum overturning (27.95 kg m$^{-3}$), approximately 60% of Dense Water produced in the Arctic Ocean originates from the Barents Sea, and approximately 40% originates from Fram Strait itself.'**

Figure 6b: I don't understand how this figure is made. Please explain.

**We have added a few sentences detailing how the Lagrangian streamfunction is calculated and decomposed in L. 276-278 and L. 280-281.**

Lines 303–304: It talks about the surface transformation occurring through cooling, melting, and/or freezing. What are the relative contributions of each of these processes?

**To assess the relative contribution of heat and freshwater fluxes we have calculated the surface water mass transformation keeping heat or freshwater fluxes at zero. This result has been added to L. 172-175 and is shown in Figure A2. The freshwater flux output from the hindcast used here only contains the net freshwater flux and not its individual contributions. Hence, we are unfortunately not able to calculate the relative contributions of melting and freezing.**

Figure 7a: I don't understand how the "overturning in density space" is calculated. Please explain.

**We added a sentence in L. 298-299 referring to the explanation given in L. 276-278, since the calculation is the same as in Figure 6b.**

Lines 350–361: The method to estimate the relative contributions of surface forcing and internal mixing to water mass transformation is ad hoc. It's hard to judge how reliable the results are, although it's reassuring to read that numbers are robust to the threshold. How can this method be tested and improved?

**We have added a sentence (L. 380-381) further stressing that this is only a rough estimate that will vary over time. One way to obtain a better estimate could be to perform a detailed regional water mass transformation analysis, but this would be beyond the scope of this work.**

Line 394: The text brushes off the apparent disagreement with the Årthun et al. (2025) paper by saying it will be a topic of another study. Meanwhile, can you speculate as to how the disagreement might be reconciled? What are the most likely explanations?

**We have expanded our discussion on the results from Årthun et al. (2025) and how they compare with ours (L. 431-442).**

Figure A1: What's the time period for the two datasets?

**In the figure caption, we now specify the period for the hindcast (1979 - 2015) and for the PHC3.0 (1950 - 2005).**

Typos etc.:

Line 101: "nu" (I think) in the integrand should be $v$ (as on line 103).

**We have fixed this.**

Line 353: "criteria" should be singular ("criterion").

**Fixed.**

**References**

Brown, N. J., Naveira Garabato, A. C., Bacon, S., Aksenov, Y., Tsubouchi, T., Green, M., Lincoln, B., Rippeth, T., and Feltham, D. L. (2025). The Arctic Ocean Double Estuary: Quantification and Forcing Mechanisms. *AGU Advances*, 6(6):e2024AV001529. _eprint: https://agupubs.onlinelibrary.wiley.com/doi/pdf/10.1029/2024AV001529.

Pemberton, P., Nilsson, J., Hieronymus, M., and Meier, H. E. M. (2015). Arctic Ocean Water Mass Transformation in S–T Coordinates. *Journal of Physical Oceanography*, 45(4):1025–1050.

Tsubouchi, T., von Appen, W.-J., Kanzow, T., and de Steur, L. (2024). Temporal Variability of the Overturning Circulation in the Arctic Ocean and the Associated Heat and Freshwater Transports during 2004–10. *Journal of Physical Oceanography*, 54(1):81–94.

Årthun, M., Brakstad, A., Dörr, J., Johnson, H. L., Mans, C., Semper, S., and Våge, K. (2025). Atlantification drives recent strengthening of the Arctic overturning circulation. *Science Advances*, 11(28):eadu1794.